# SIRT4 as a novel interactor and candidate suppressor of C-RAF kinase in MAPK signaling

Mehrnaz Mehrabipour[1], Saeideh Nakhaei-Rad[2] , Radovan Dvorsky[1] , Alexander Lang[1], Patrick Verhülsdonk[1], Mohammad R Ahmadian[1,*] , Roland P Piekorz[1,*]

Cellular responses leading to development, proliferation, and differentiation depend on RAF/MEK/ERK signaling, which integrates and amplifies signals from various stimuli for downstream cellular responses. C-RAF activation has been reported in many types of tumor cell proliferation and developmental disorders, necessitating the discovery of potential C-RAF protein regulators. Here, we identify a novel and specific protein interaction between C-RAF among the RAF kinase paralogs, and SIRT4 among the mitochondrial sirtuin family members SIRT3, SIRT4, and SIRT5. Structurally, C-RAF binds to SIRT4 through the N-terminal cysteine-rich domain, whereas SIRT4 predominantly requires the C-terminus for full interaction with C-RAF. Interestingly, SIRT4 specifically interacts with C-RAF in a pre-signaling inactive (serine 259–phosphorylated) state. Consistent with this finding, the expression of SIRT4 in HEK293 cells results in an up-regulation of pS259-C-RAF levels and a concomitant reduction in MAPK signaling as evidenced by strongly decreased phospho-ERK signals. Thus, we propose an additional extra-mitochondrial function of SIRT4 as a cytosolic tumor suppressor of C-RAF-MAPK signaling, besides its metabolic tumor suppressor role of glutamate dehydrogenase and glutamate levels in mitochondria.

## Introduction

C-RAF (often also called RAF1) belongs to the RAF kinase family (A-RAF, B-RAF, and C-RAF), which transfers proliferative and growth signals to downstream activation of MEK/ERK kinases. These RAF paralogs share several structural properties (Rezaei Adariani et al, 2018; Nakhaei-Rad et al, 2023b), yet they differ in terms of activity levels and functional roles (Desideri et al, 2015). Among them, C-RAF exhibits moderate activity, less than B-RAF, but more than A-RAF, and is associated with cancer and developmental disorders (Blasco

et al, 2011; Karreth et al, 2011; Gelb et al, 2015; Degirmenci et al, 2020). There are three conserved regions (CR) within RAF proteins that are important for their respective regulatory functions (CR1 and CR2) and kinase activity (CR3) (Rezaei Adariani et al, 2018). CR1 contains a RAS-binding domain (RBD), mediating a RAS interaction, and a cysteine-rich domain (CRD), which mediates membrane binding and enhances RAS/RBD affinity at the membrane (Fang et al, 2020; Tran et al, 2021; Nguyen et al, 2022). CR2 is enriched by several Ser/Thr residues, including serine 259 (S259), which is an important site for inhibitory phosphorylation and 14-3-3 binding that regulates RAF kinase activation (Dhillon et al, 2002). When phosphorylated by upstream kinases such as AKT, PKA, or LATS1, CR2 acts as an inhibitory domain that keeps C-RAF in an inactive state (Zimmermann & Moelling, 1999; Dumaz & Marais, 2003; Romano et al, 2014). Dephosphorylation of CR2 by protein phosphatases, such as PP2A or PP1, relieves this autoinhibition on the kinase domain and activates C-RAF (Jaumot & Hancock, 2001). CR3 functions as a catalytic C-terminal region, constituting a putative phosphorylation segment for kinase activation (Chong et al, 2001). Thus, C-RAF cycles between a close inactive and an open active conformation, which is regulated by different phosphorylation and dephosphorylation events (Lavoie & Therrien, 2015). Overall, phosphorylation, feedback/autoinhibition, and protein–protein interaction occur in C-RAF regulation in response to signaling events (Wimmer & Baccarini, 2010; Cseh et al, 2014; Romano et al, 2014; Lavoie & Therrien, 2015; Varga et al, 2017; Okamoto & Sako, 2023). In particular, RAS and 14-3-3 binding are major regulatory events of RAF activation, membrane recruitment, and stabilization (Matallanas et al, 2011; Li et al, 2018; Jang et al, 2020; Tran et al, 2021). Addressing the molecular control of C-RAF by interacting regulators and the underlying molecular and structural mechanisms is still necessary for understanding the complex landscape of MAPK network signaling. Several proteins that bind and regulate C-RAF have been identified, including RKIP (RAF1 kinase inhibitor protein), which functions as an anti-metastatic tumor suppressor and is down-regulated in various cancers (Yesilkanal & Rosner, 2018; Touboul et al, 2021; Cessna et al,

[1]Institute of Biochemistry and Molecular Biology II, Medical Faculty and University Hospital Düsseldorf, Heinrich Heine University, Düsseldorf, Germany   [2]Stem Cell Biology, and Regenerative Medicine Research Group, Institute of Biotechnology, Ferdowsi University of Mashhad, Mashhad, Iran

Correspondence: Reza.Ahmadian@hhu.de; Roland.Piekorz@hhu.de
Alexander Lang's present address is Department of Cardiology, Pulmonology, and Vascular Medicine, Medical Faculty and University Hospital Düsseldorf, Heinrich Heine University, Düsseldorf, Germany
*Mohammad R Ahmadian and Roland P Piekorz contributed equally to this work

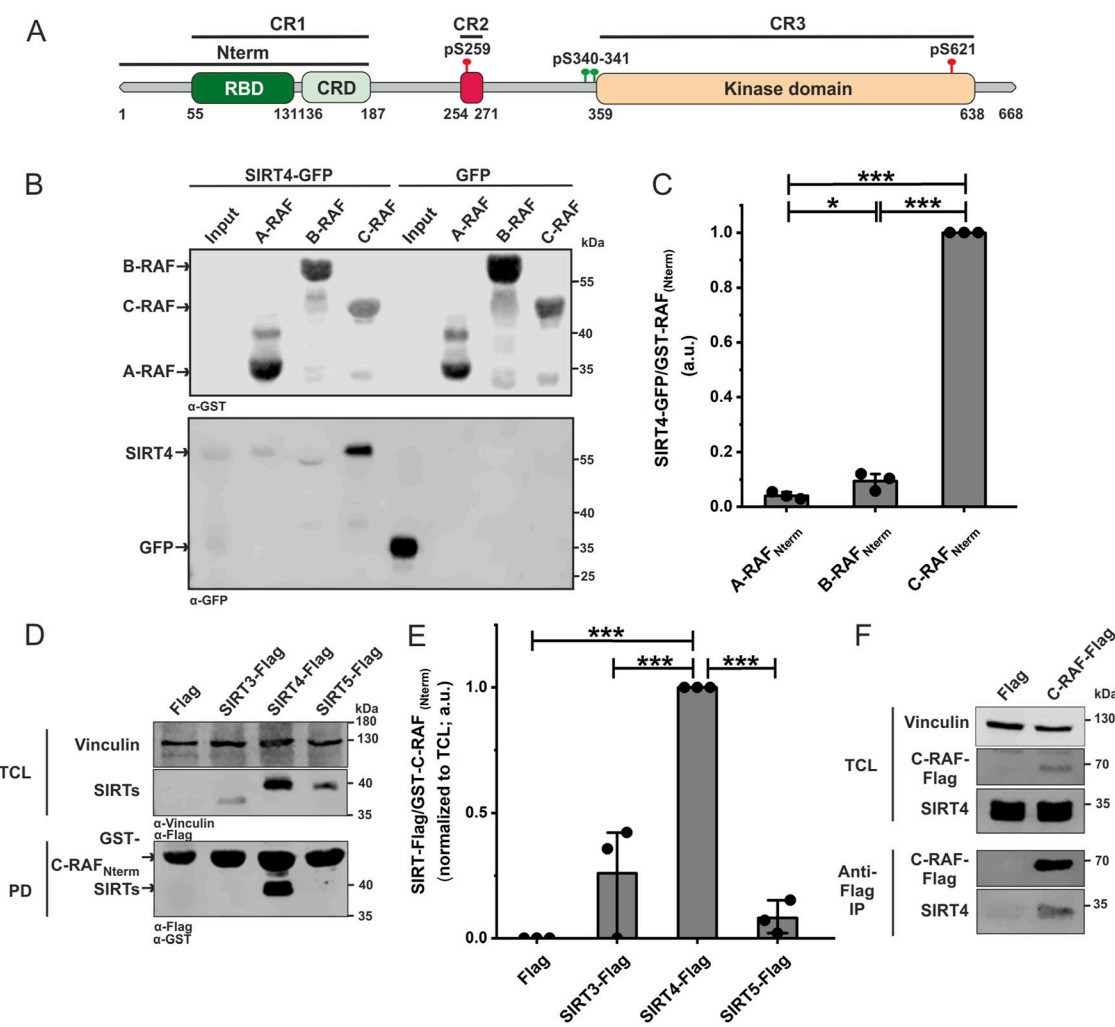

**Figure 1. Identification of a selective interaction between SIRT4 and C-RAF within the RAF kinase and SIRT paralogs.**
**(A)** Domain organization of C-RAF including the RAS-binding domain and cysteine-rich domain, which are parts of the N-terminal region (Nterm). Phosphorylation sites regulating the activity of C-RAF (pS259: inactive form; pY340/341: active form) are indicated. **(B)** Total cell lysates (TCL) from SIRT4-eGFP– or eGFP-expressing HEK293 cells were subjected to pull-down experiments using normalized bacterial lysates containing the GST-fused Nterm region of A-RAF, B-RAF, or C-RAF. **(C)** Densitometric quantification of immunoblot signals of binding of SIRT4-eGFP to the N-RBD-CRD of C-RAF as compared to A-RAF and B-RAF. Data were subjected to statistical one-way ANOVA (mean ± S.D.; *$P < 0.05$; ***$P < 0.001$). **(D)** TCL from HEK293 cells expressing Flag-tagged versions of SIRT3, SIRT4, or SIRT5 were subjected to pull-down (PD) experiments using the GST-fused Nterm region of C-RAF. **(E)** Densitometric quantification of immunoblot signals of binding of the Nterm region of C-RAF to SIRT4 as compared to SIRT3 and SIRT5. Data were subjected to statistical one-way ANOVA (mean ± S.D.; ***$P < 0.001$). **(F)** Co-immunoprecipitation analysis (anti-Flag Co-IP) of endogenous SIRT4 was performed using TCL from Flag-C-RAF–expressing COS7 cells.
Source data are available for this figure.

2022; Moghaddam et al, 2023). RKIP binds to the N-terminal region of C-RAF and therefore inhibits C-RAF–mediated phosphorylation and activation of MEK1/2 (Rath et al, 2008).

The family of human sirtuins comprises seven members, of which SIRT3, SIRT4, and SIRT5 function as bona fide metabolic regulators in mitochondria (Ji et al, 2022). In particular, SIRT4 inhibits, as a tumor suppressor, the metabolic gatekeepers pyruvate dehydrogenase and glutamate dehydrogenase (Haigis et al, 2006; Mathias et al, 2014), with particular significance for the regulation of glutamine metabolism in tumor cells. Recent findings uncovered novel extra-mitochondrial roles of SIRT4 in microtubule dynamics and regulation of mitotic cell cycle progression, WNT/β-catenin and Hippo signaling, and SNARE complex formation required for

autophagosome–lysosome fusion (Bergmann et al, 2020; Wang et al, 2022; Yang et al, 2022; Huang et al, 2023). Interestingly, proteomic analysis of the SIRT4 interactome identified C-RAF as a potential binding partner of SIRT4, indicating a novel role of SIRT4 in the regulation of the RAF-MAPK signaling pathway (Bergmann et al, 2020). Consistent with this idea, recent studies have demonstrated that (i) the tumor suppressor SIRT4 is down-regulated in most human solid tumor types and cell lines (Bai et al, 2020; Tomaselli et al, 2020; Wang et al, 2020), and (ii) the ectopic expression of SIRT4 down-regulates pERK1/2 levels and hence inhibits MAPK signaling and cell proliferation (Fu et al, 2017; Chen et al, 2019; Hu et al, 2019; Bai et al, 2020; Tomaselli et al, 2020; Wang et al, 2020). Considering these interrelated findings, in this study we investigated the

molecular and functional interaction between the proto-oncogene C-RAF and the tumor suppressor SIRT4 in the context of MAPK signaling inhibition.

# Results

### Identification of a selective SIRT4-C-RAF interaction among SIRT and RAF protein family members

In a previous study, we employed mass spectrometry and proteomic analysis to identify novel SIRT4-interacting proteins (Bergmann et al, 2020). Interestingly, C-RAF kinase (often referred to by its gene name *Raf1*), a major component of the MAPK signaling pathway, emerged as a novel SIRT4-binding protein as confirmed by nanobody-based co-immunoprecipitation analysis (Fig S1). Considering the presence of N-terminal regulatory (CR1, CR2) and C-terminal catalytic (CR3) domains in C-RAF (Fig 1A), we hypothesized that the N-terminal CR1 regulatory segment, consisting of the RBD (RAS-binding domain) and CRD, might be involved in SIRT4 interaction.

Accordingly, we addressed the specificity of SIRT4-C-RAF interaction by protein pull-down analysis using bacterially expressed GST-fused N-terminal (Nterm) regions of A-RAF, B-RAF, or C-RAF, each containing the respective RBD and CRD. Normalized amounts of GST-RAF lysates were coupled to GSH (glutathione) beads followed by incubation with total cell lysates from HEK293 cells expressing SIRT4-GFP or GFP as a control. As indicated in Figs 1B and C and S2A, a strong physical interaction with SIRT4 was only observed for C-RAF$_{Nterm}$, but not for A-RAF$_{Nterm}$ or B-RAF$_{Nterm}$. In complementary pull-down experiments, we used total cell lysates from HEK293 cells stably expressing C-terminally Flag-tagged SIRT3, SIRT4, or SIRT5. Only SIRT4 exhibited a robust interaction with C-RAF$_{Nterm}$, but not SIRT3 or SIRT5 (Figs 1D and E and S2B). Finally, we immunoprecipitated Flag-tagged C-RAF from COS7 cell lysates and could demonstrate co-immunoprecipitation of endogenous SIRT4 (Figs 1F and S2C). Overall, our data suggest that within the sirtuin and RAF family members studied, only C-RAF and SIRT4 undergo a unique interaction.

### The CRD of C-RAF and the C-terminus of SIRT4 are major determinants of the interaction between SIRT4 and C-RAF

In the next step, we sought to determine the regions or subdomains of C-RAF and SIRT4 that are directly involved in the interaction between these two proteins. We expressed GST-C-RAF-Nterm, RBD, and CRD in *Escherichia coli* and used them to pull down SIRT4-Flag from total cell lysates of HEK293 cells. As indicated in Fig S3A–C, C-RAF$_{Nterm}$ and interestingly CRD alone (C-RAF$_{CRD}$) bound to SIRT4-Flag, although with a higher efficiency seen for C-RAF$_{Nterm}$. However, no or only a slight interaction with SIRT4-Flag could be observed for the RBD (C-RAF$_{RBD}$) (Figs 2A and B and S3A–D). These results suggest that the CRD is the major SIRT4-binding domain of C-RAF.

In order to get insight into molecular aspects of SIRT4 binding to C-RAF, we set out to inspect the structures of these proteins and analyze their putative complex. We first generated a homology model of human SIRT4 using the 3D structure of SIRT4 from *Xenopus tropicalis* (PDB: 5OJ7) (Pannek et al, 2017) as a template. Given that SIRT4, but neither SIRT3 nor SIRT5, binds to C-RAF$_{Nterm}$ (Fig 1B and C), we have scrutinized their sequences and compared our model structure of SIRT4 with the structure of human SIRT5 (PDB: 4G1C) (Fig S4A and B). This analysis revealed three regions in SIRT4 that differ from SIRT5, that is, R1$_{(69–98)}$, R2$_{(165–229)}$, and R3$_{(255–314)}$ (Figs 2C and D and S4). The corresponding SIRT4 deletion mutants SIRT4(Δ69–98; ΔR1), SIRT4(Δ165–229; ΔR2), and SIRT4(Δ255–314; ΔR3) were generated as C-terminal GFP-tagged proteins, stably expressed in HEK293 cells, and tested for C-RAF$_{Nterm}$ binding in pull-down experiments. As shown in Figs 2E and F and S3E, SIRT4(ΔR3) strikingly showed the weakest interaction with C-RAF$_{Nterm}$, whereas ΔR1 and ΔR2 were not significantly different from wild-type SIRT4. Moreover, SIRT4(ΔN28), which lacks the N-terminal mitochondrial translocation signal (Lang et al, 2017), as well as the catalytically inactive mutant SIRT4(H161Y) (Lang et al, 2017), bound C-RAF$_{Nterm}$ comparable to WT SIRT4 (Figs 2E and F and S5A–C). Taken together, C-RAF$_{CRD}$ and the C-terminus of SIRT4, encompassing residues 255–314, are involved in SIRT4-C-RAF interaction, which is independent of the first 28 a.a. of SIRT4 and therefore its mitochondrial localization and of the catalytic activity of SIRT4. Our findings also add a new function to the C-terminus of SIRT4 besides its role in proteasomal degradation and stability regulation of SIRT4 (Hampel et al, 2023).

### Mutational analysis of the interaction between C-RAF$_{CRD}$ and SIRT4

We generated nine single mutations and three sets of combined mutations of C-RAF$_{CRD}$ based on the multiple sequence alignment of amino acid deviations of C-RAF$_{CRD}$ in comparison with the CRD of A-RAF and B-RAF (Fig 3A and B). All mutants were expressed and purified as GST-fusion proteins and subjected to pull-down assays using total cell lysates from SIRT4-Flag–expressing HEK293 cells. As indicated in Figs 3C and E and S6, and quantitatively analyzed in Fig 3D and F, none of the single or combined mutants analyzed a decreased interaction of C-RAF$_{CRD}$ with SIRT4-Flag. Rather, we observed significantly stronger binding for the CRD mutants Q156R, Set1 (E174Q/H175R/T178S/K179E/T182L), and Set2 (Q156R/F158L/L160F) (Fig 3C–F).

To identify residues of the C-RAF$_{CRD}$-SIRT4-binding interface and obtain a more detailed insight into their intermolecular interplay, we performed molecular docking analysis between C-RAF$_{CRD}$ (PDB: 1FAQ) and full-length SIRT4 (Q9Y6E7) using the ClusPro 2.0 server. The 3D surface structure (Fig 3G) highlights the binding between C-RAF$_{CRD}$ and R3 of SIRT4, along with certain parts of R1. For a more detailed understanding of this intermolecular binding, analysis of the binding surface using BIOVIA software revealed an interacting network (Fig 3H), in which the stability of the C-RAF$_{CRD}$-SIRT4 complex is the result of a combination of various interaction types, that is, hydrogen bonds, electrostatic interactions, and hydrophobic contacts (Table S1). For example, the C-RAF$_{CRD}$ residue K157 and the SIRT4 residue D236 form a hydrogen/electrostatic bond with a distance of 1.8 Å, indicative of a strong interaction. C-RAF$_{CRD}$ residues R143, K157, H175, T178, K179, Q156, E174, S177, N161, and I154, and SIRT4 residues R75, R97, T274, H92, T237, D236, Q264, Q91, R270, R291, G93, G235, and Y266 further contribute to the binding stability

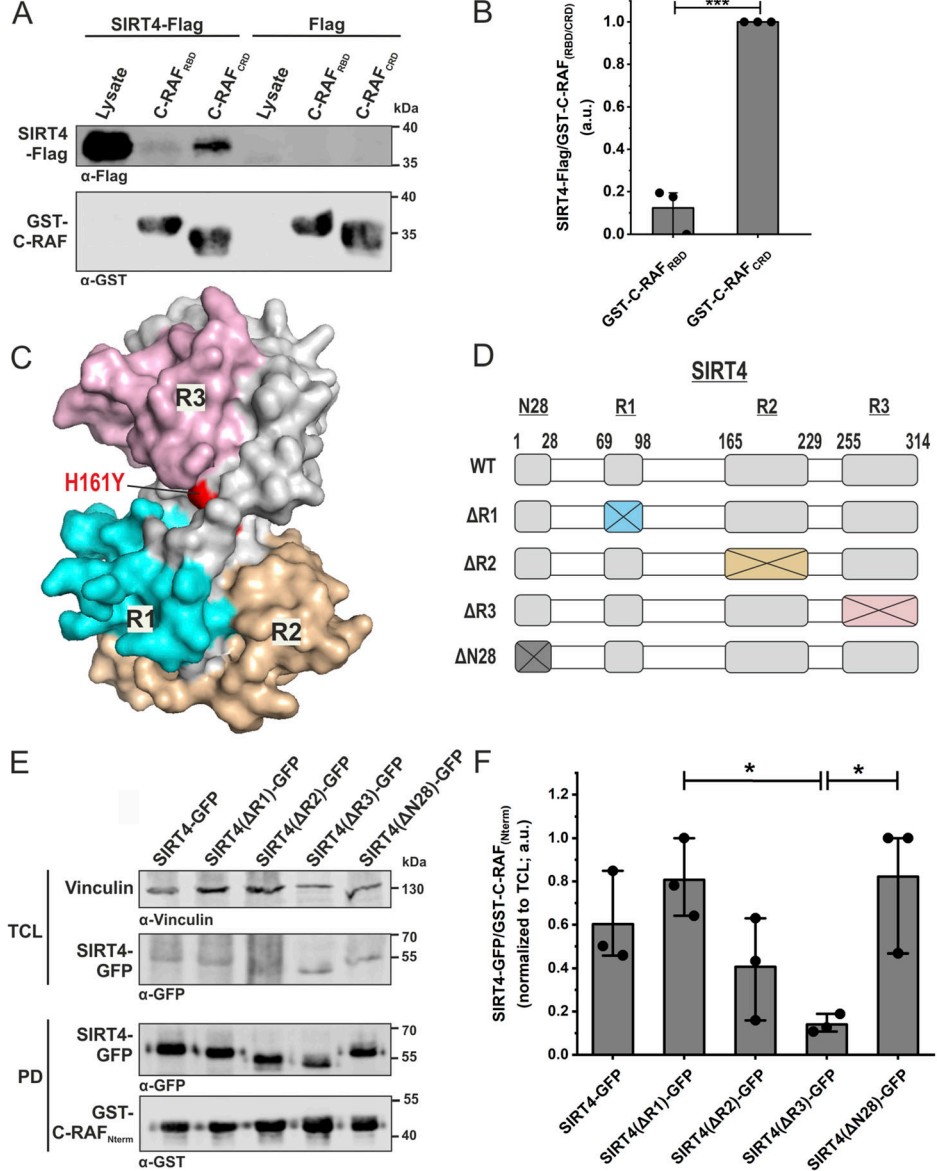

**Figure 2. Identification of a selective interaction between the cysteine-rich domain of C-RAF and the very C-terminal region of SIRT4.**
**(A)** Identification of the CRD of C-RAF as the primary SIRT4-interacting domain. Total cell lysates from HEK293 cells stably expressing SIRT4-Flag were subjected to pull-down experiments using GST or the GST-fused N-terminal RBD or CRD subdomains of C-RAF. **(B)** Densitometric quantification of immunoblot signals of the relative binding of RBD and CRD subdomains of C-RAF to SIRT4-Flag. Data were subjected to statistical one-way ANOVA (mean ± S.D.; \*\*\*$P < 0.001$). **(C)** Predicted functional surface of SIRT4 was obtained from comparative homology modeling with SIRT5 (see Fig S4A and B). Three regions (R1, R2, and R3), which are different between SIRT4 and SIRT5, are highlighted in the 3D-modeled SIRT4 structure. Replacement of histidine 161 by tyrosine creates the catalytically inactive SIRT4. **(D)** Schematic representation of SIRT4 deletion mutants, including ΔR1, ΔR2, ΔR3, and ΔN28 lacking the N-terminal mitochondrial translocation sequence. **(E)** Equal amounts of total cell lysates from HEK293 cells expressing the SIRT4-eGFP of the indicated deletion mutants were subjected to pull-down (PD) analysis using the GST-fused C-RAF_Nterm_. **(F)** Densitometric quantification of immunoblot signals of the relative binding of SIRT4-GFP deletion mutants to the GST-fused C-RAF_Nterm_. Data were subjected to statistical one-way ANOVA (mean ± S.D.; \*$P < 0.05$).
Source data are available for this figure.

via hydrogen bonds. Notably, electrostatic interactions were observed between C-RAF_CRD_ residues R143, E174, and F141, and SIRT4 residues E277, R270, and R291, respectively (Fig 3H; Table S1). Moreover, hydrophobic interactions were identified involving residues of C-RAF_CRD_ (H175, L160, F163, R143) and SIRT4 (V232, F234, P240, Y266, R270).

Because the C-RAF_CRD_ Set1 and Set2 mutations resulted in stronger binding to SIRT4-Flag (Fig 3C–F), further molecular docking analysis was performed for these C-RAF_CRD_ gain-of-function mutations. Comparing the cluster scores of WT C-RAF_CRD_ interacting with SIRT4 shows a weighted score of –716 for both the middle and the lowest energy. In contrast, Set1 and Set2 have lower, more stable cluster scores: –738.7 and –795 for the center and the lowest energy in the case of Set1, and –744 for both the center and the lowest energy in the case of Set2. The combined mutations in Set1,

particularly the E174Q, H175R, T175S, K179E, and T182L mutations, alter the interaction profile of C-RAF_CRD_ with SIRT4, thereby forming new hydrogen bonds, as well as electrostatic and hydrophobic contacts, which potentially enhance complex stability (Fig S7D and Table S2). Although some interactions are lost in Set1 compared with WT C-RAF_CRD_ (Table S2), considering the cluster score and the mode of binding, we propose also new platforms of interactions. These involve a new set of C-RAF_CRD_ residues, that is, D153, Y170, P181, L182, M183, and V185, that might collectively increase the binding affinity of Set1 to SIRT4 (Fig S7D and Table S2). Moreover, compared with WT C-RAF_CRD_, the mutations within Set1 induce a modified interaction pattern with SIRT4, characterized by an increased interaction of C-RAF_CRD_ residues with R1 of SIRT4 while exhibiting a reduced interaction with R3 and the SIRT4 gray area (which lacks R1, R2, and R3) (Fig S7A–D).

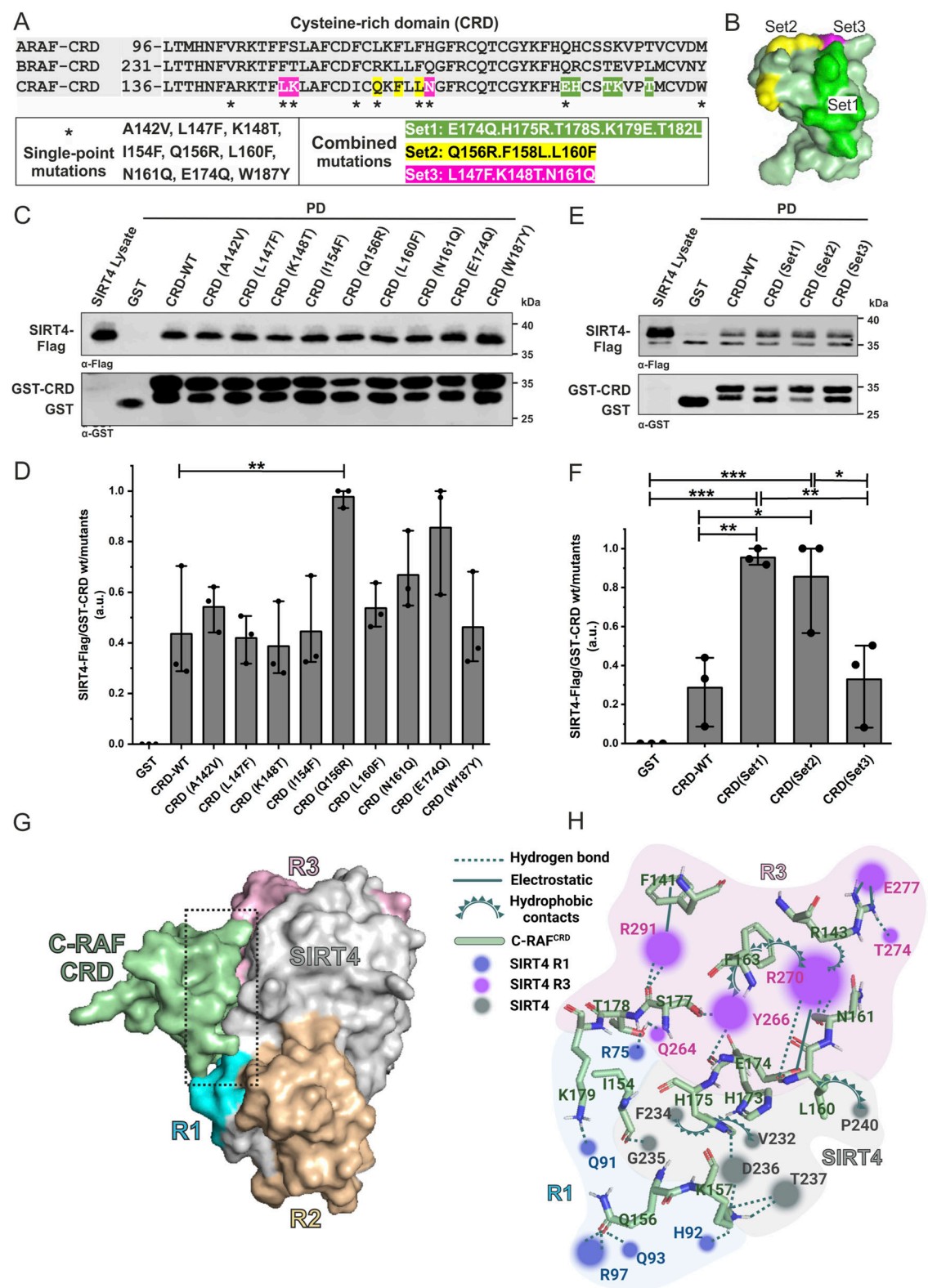

**Figure 3. Mapping the SIRT4-binding site of C-RAF.**
**(A)** Multiple sequence alignment highlights amino acid deviations of the CRD of C-RAF as compared to the CRD of A-RAF and B-RAF and is the basis for single-point and combined mutations of C-RAF generated in this study. **(B)** 3D model of the three sets of combined mutations in the CRD of C-RAF. **(A, C, E)** Total cell lysates from HEK293 cells stably expressing SIRT4-Flag were subjected to pull-down experiments using GST, GST-CRD (WT), or GST-CRD harboring single-point mutations (C) or combined mutations (E) as indicated in (A). **(D, F)** Densitometric quantification of immunoblot signals of the relative binding of WT and mutated CRD subdomains of C-RAF to

Similar to C-RAF$_{CRDSet1}$, Set2 mutations in the C-RAF$_{CRD}$ region also introduce new interactions, as well as changes in the type and distance of existing interactions with the respective SIRT4 regions (Fig S7E and F). For instance, the F158L mutation leads to the formation of a new hydrogen bond with T237 of SIRT4, and the L160F mutation results in the interaction with both P240 and V243 of SIRT4, leading to a higher involvement of CRD-Set2 residues (Fig S7F and Table S2). Notably, in the case of C-RAF$_{CRDSet2}$, the C-RAF$_{CRD}$ residues C155, L158, F172, and H173 undergo novel hydrogen bonds with SIRT4 residues, suggesting a restructuring of the binding interface and thereby increasing the stability of the C-RAF$_{CRD}$-SIRT4 interaction in the case of C-RAF$_{CRDSet2}$ (Fig S7F and Table S2).

### SIRT4 binds selectively to the inactive state of C-RAF characterized by phosphorylation of serine 259

C-RAF exists in two distinct forms. Its closed, monomeric, autoinhibited form is stabilized by phosphorylation at serines 259 and 621 (pS259/pS621), and subsequent association with the 14-3-3 dimer (Rommel et al, 1996; Matallanas et al, 2011). The C-RAF activation involves a series of complex processes, including dephosphorylation (pS259) and phosphorylation (pY340/pY341) events, conformational changes, dimerization, and association with RAS, 14-3-3, and the membrane, ultimately stabilizing the open, dimeric, active form of C-RAF (Emerson et al, 1995; Diaz et al, 1997; Jaumot & Hancock, 2001; Harding et al, 2003; Terai & Matsuda, 2005; Takahashi et al, 2017). Thus, we addressed whether SIRT4 interacts with C-RAF in its active or inactive state. As indicated in Figs 4A and S8A, endogenously expressed C-RAF could be immunoprecipitated from total cell lysates of HEK293 cells expressing SIRT4-Flag. However, when using specific antibodies against pS259-C-RAF (closed, inactive form) and pY340/341-C-RAF (open, active form), only pS259-C-RAF was detected in the immunoprecipitates (Fig 4A). These findings are consistent with homology modeling of C-RAF$_{CRD}$ in the inactive form of C-RAF (Fig 4B), in which the putative SIRT4-binding region remains accessible as part of the C-RAF$_{CRD}$ domain (indicated in pale green). Furthermore, co-immunoprecipitation of KRAS within the SIRT4-Flag-C-RAF–interacting complex could not be detected (Figs 4A and S8A), supporting the notion that C-RAF exclusively exists in its autoinhibited form in complex with SIRT4. Overall, this is consistent with an interaction of KRAS only with the active form of C-RAF, which requires dephosphorylation of S259 and unmasking of the RBD and CRD to allow KRAS binding to C-RAF at the membrane (reviewed in Matallanas et al [2011]). Further structural analysis provides additional evidence that the SIRT4-binding region of C-RAF$_{CRD}$ contains residues required for KRAS–membrane interaction (Fig 4B).

### SIRT4-C-RAF interaction is associated with the inhibition of the MAPK signaling pathway

It is well established in the literature that SIRT4 overexpression inhibits cell proliferation, among other cellular responses, in several tumor cell lines, most likely through inhibition of the MAPK pathway (Fu et al, 2017; Chen et al, 2019; Hu et al, 2019; Bai et al, 2020; Tomaselli et al, 2020; Wang et al, 2020). Here, we addressed the regulatory affect of ectopic SIRT4 expression on ERK1/2 phosphorylation. As shown in Figs 4C and D and S8B, the ectopic expression of SIRT4 led to a clear accumulation of the levels of inactive C-RAF phosphorylated at S259. At the same time, MAPK signaling was strongly inhibited as evidenced by an ~80% reduction in p-ERK1/2 levels as compared to Flag-expressing control cells. Overall, these data suggest that SIRT4 both interacts with and possibly sequesters the inactive form of C-RAF. Thus, the extra-mitochondrial function of SIRT4 on C-RAF-MAPK signaling may provide a novel control mechanism for tumor suppression (Fig 4E).

## Discussion

The work presented in this study has identified a novel interaction of SIRT4, a tumor suppressor sirtuin, with C-RAF, a key regulatory kinase and a component of the oncogenic MAPK pathway. The results indicate that (i) among the RAF kinases (A-RAF, B-RAF, and C-RAF) and sirtuin proteins (SIRT3, SIRT4, and SIRT5) analyzed, C-RAF selectively interacts with SIRT4; (ii) this interaction involves the N-terminal CRD of C-RAF and the C-terminal region 3 (R3) of SIRT4 as revealed by pull-down and molecular docking analyses; (iii) mutational analysis of C-RAF$_{CRD}$ so far identified gain-of-function mutations with improved SIRT4 binding, thus highlighting the importance of these residues in the C-RAF$_{CRD}$-SIRT4 interaction; (iv) in particular, SIRT4 specifically interacts with C-RAF in its inactive state (C-RAF$^{pS259}$); and (v) the ectopic expression of functional SIRT4 leads to accumulation of pS259-C-RAF levels, which is associated with inhibition of MAPK signaling as shown by greatly reduced p-ERK1/2 levels. Thus, our data highlight a novel extra-mitochondrial, anti-proliferative function of SIRT4 in binding and potentially sequestering C-RAF from its substrate MEK1/2 and consequently interfering with ERK1/2 activation.

The MAPK signaling pathway plays a critical role in the regulation of various cellular processes such as differentiation, survival, and, in particular, proliferation (Zhang & Liu, 2002; Guo et al, 2020; Ullah et al, 2022). Dysregulation of this pathway is frequently associated with the initiation and progression of human diseases, including cancer (Degirmenci et al, 2020) and developmental disorders such as RASopathies (Dar & Brady, 2022), the latter exemplified by the RAF1$^{S257L}$ mutation causing cardiomyopathy (Dhandapany

SIRT4-Flag. Data were subjected to statistical one-way ANOVA (mean ± S.D.; *P < 0.05; **P < 0.01; ***P < 0.001). **(G)** Molecular docking and binding site analyses between the CRD of C-RAF and specified regions of SIRT4. The predicted interaction between the CRD of C-RAF and the C-terminal region R3 of SIRT4, along with a smaller part of R1, is depicted in this 3D model. **(H)** Schematic, magnified view of the CRD-SIRT4–interacting surface and the involved amino acid residues. The binding types, that is, hydrogen bonds, electrostatic interactions, and hydrophobic contacts, are indicated. The colored circles mark SIRT4 residues, with the size of the circles indicating the number of interactions with the CRD.
Source data are available for this figure.

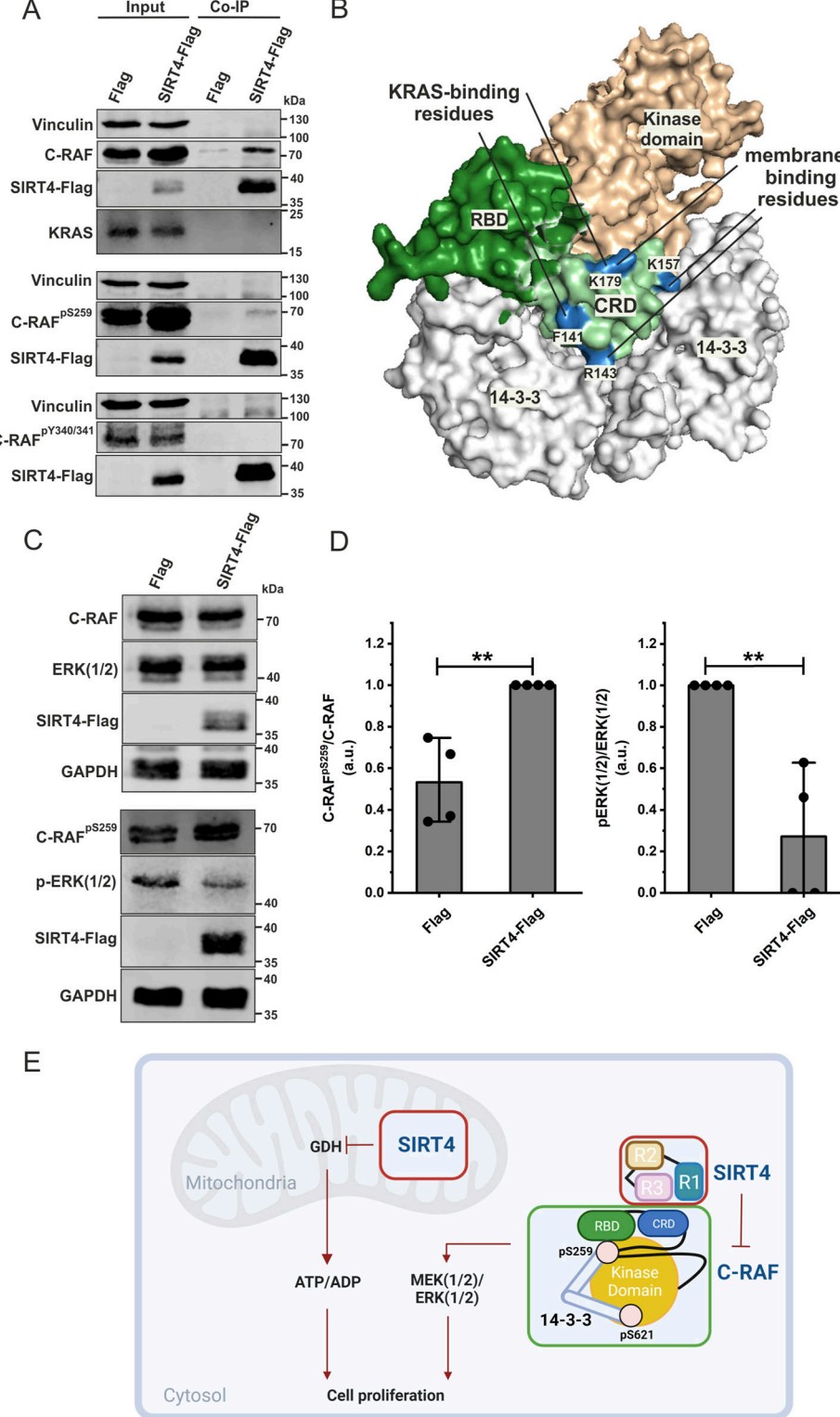

**Figure 4.   SIRT4 interacts with and up-regulates the inactive form of C-RAF phosphorylated at serine 259 (S259).**
**(A)** Co-immunoprecipitation analysis using total cell lysates (Input) from HEK293 cells expressing Flag or SIRT4-Flag shows the SIRT4 interaction specifically with C-RAF in its autoinhibited state (pS259-C-RAF) but not with pY340/341-C-RAF in its active state. Moreover, KRAS did not co-immunoprecipitate with the SIRT4-pS259-C-RAF complex. **(B)** Homology model of the closed, inactive C-RAF structure in complex with the 14-3-3 dimer (light gray) was built using the crystal structure of B-RAF as a template. The accessibility of the CRD in its inactive form is represented (pale green). The model depicts regions highlighted in blue that are crucial for KRAS binding and membrane interaction in the active state of C-RAF. The amino acids involved are indicated. **(C)** Total cell lysates from HEK293 cells expressing Flag or SIRT4-Flag were subjected to immunoblot analysis of pS259-C-RAF and pERK1/2 levels. The ectopic expression of SIRT4 in HEK293 cells increased the levels of inactive pS259-C-RAF and reduced ERK1/2 phosphorylation. **(D)** Densitometric immunoblot analysis of the levels of pS259-C-RAF (left panel) and pERK1/2 (right panel) upon Flag or SIRT4-Flag expression was subjected to statistical one-way ANOVA (mean ± S.D.; **$P$ < 0.01). **(E)** Hypothetical model summarizing the two anti-proliferative axes of SIRT4. SIRT4 displays bifunctional activities in inhibiting glutamate dehydrogenase in mitochondria and C-RAF-MAPK signaling in the cytosol. For further explanation, see Discussion.
Source data are available for this figure.

et al, 2014; Jaffre et al, 2019; Nakhaei-Rad et al, 2023a). As a key component of the MAPK pathway, C-RAF is activated by upstream receptor–RAS signaling and subsequently activates several downstream effectors, particularly MEK1/2 kinases and subsequently

ERK1/2 signaling (Wimmer & Baccarini, 2010; Matallanas et al, 2011; Ullah et al, 2022). Several studies have highlighted the molecular mechanism of C-RAF regulation underlying post-translational modifications through phosphorylation and dephosphorylation,

autoinhibition, and conformational changes associated with stabilized protein–protein interaction (Romano et al, 2014; Lavoie & Therrien, 2015; Varga et al, 2017; Okamoto & Sako, 2023). Classically, RAS proteins and 14-3-3 binding are major regulators of RAF activation, membrane recruitment of C-RAF, and its stability (Matallanas et al, 2011; Tran et al, 2021). The complexity of C-RAF regulation is further highlighted by its heterodimerization with B-RAF, which acts as an allosteric inducer of C-RAF in normal and cancer cells in a RAS-independent manner (Garnett et al, 2005).

Recent findings have identified and characterized additional C-RAF regulators. SHOC2 serves as a scaffold protein for C-RAF that recruits together with MRAS protein phosphatase 1 to dephosphorylate inactive C-RAF at S259, thereby facilitating the C-RAF interaction with RAS at the plasma membrane (Matsunaga-Udagawa et al, 2010; Boned Del Rio et al, 2019; Kwon et al, 2022). In another example, SHOC2 serves as a regulatory factor for C-RAF and has been shown to accelerate the interaction between RAS and C-RAF, ultimately influencing the spatiotemporal patterns of the RAS-ERK signaling pathway (Matsunaga-Udagawa et al, 2010). Moreover, RKTG (RAF kinase trapping to Golgi) has been suggested to regulate the spatial localization of C-RAF by trapping it to the Golgi, thereby altering the interaction of C-RAF with RAS and MEK1 and inhibiting ERK signaling (Feng et al, 2007). Another regulator of C-RAF is RKIP (Yesilkanal & Rosner, 2018; Touboul et al, 2021; Cessna et al, 2022; Moghaddam et al, 2023), which binds to the N-terminal region of C-RAF, thereby inhibiting C-RAF–mediated phosphorylation and activation of MEK1/2 (Park et al, 2006; Rath et al, 2008). Interestingly, a comparison between RKIP and SIRT4 reveals cellular and functional similarities: (i) both proteins are tumor suppressors (Jeong et al, 2013; Moghaddam et al, 2023) that inhibit/prevent C-RAF activation, and their expression is usually down-regulated in cancer (Yesilkanal & Rosner, 2018; Bai et al, 2020; Tomaselli et al, 2020; Wang et al, 2020), although the underlying mechanisms for SIRT4 are still unclear; (ii) SIRT4 and RKIP are both involved in the regulation of mitotic cell division. SIRT4 achieves this through centrosomal localization and potential control of microtubule dynamics (Bergmann et al, 2020), whereas RKIP achieves this through interaction with Aurora-B and control of the mitotic checkpoint (Eves et al, 2006); and finally, (iii) both SIRT4 (Lang et al, 2017; Li et al, 2023) and RKIP are linked to the regulation of autophagy. RKIP is involved in LC3 processing and presumably contributes to autophagosome formation upon starvation (Noh et al, 2016; Wang & Bonavida, 2018). The role of the SIRT4-C-RAF axis in the regulation of these cellular responses requires further characterization.

Interestingly, analogous to our finding, the role of C-RAF$_{CRD}$ interaction in an isoform-specific manner with another C-RAF regulator to inhibit the MAPK pathway has been demonstrated for RAP1 (Nussinov et al, 2020). Here, RAP1 inhibits MAPK signaling via interaction with C-RAF$_{CRD}$ by reducing the number of clustered oncogenic Ras molecules, thereby suppressing C-RAF (but not B-RAF) activation and MAPK signaling. The presence of RAP1 within the nanoclusters competes with RAS for C-RAF as a common target, resulting in the suppression of C-RAF activation. However, whereas RAP1 interacts with the open form of C-RAF at the cell membrane, our data suggest that SIRT4 binds to the autoinhibited (closed) form

of C-RAF. Regardless, similar to RAP1, SIRT4 may functionally hijack and inhibit C-RAF via its CRD.

The intermolecular interplay within the C-RAF$_{CRD}$-SIRT4-binding interface remains to be determined at the residual level. The single and combined C-RAF$_{CRD}$ mutations, defined by homology comparison with the CRD of A-RAF and B-RAF (which do not interact with SIRT4), did not negatively interfere with the C-RAF$_{CRD}$-SIRT4 interaction (Fig 3). Therefore, molecular docking experiments of C-RAF$_{CRD}$ on SIRT4 were performed to determine their putative binding interface. In addition to the residues identified in the mutational analysis of the C-RAF$_{CRD}$ domain (Fig 3), these analyses revealed other candidate residues that may be critical for the interaction with SIRT4 (Fig 3H and Table S1). In addition, candidate residues within the R3 and R1 regions of SIRT4 were identified, whose function also needs to be tested by mutational analysis.

Interestingly, the SIRT4-binding region of C-RAF$_{CRD}$ contains residues that are also required for KRAS and membrane interaction of C-RAF$_{CRD}$ (Fig 4B). Previous results identified seven essential basic residues within the CRD (R143, K144, K148, K157, R164, K171, and K179) that are critical for membrane interaction, with particular emphasis on the key basic residues R143, K144, and K148 (Li et al, 2018). R143, K157, and K179 are accessible in the inactive state of C-RAF and are part of the SIRT4 interaction surface, whereas the remaining residues are located on the opposite side and are shielded by 14-3-3 dimers (Fig 4B). In terms of KRAS binding, F141 and K179 are critical for the interaction between KRAS and C-RAF during the activation process (Tran et al, 2021). In the inactive state of C-RAF, in addition to K179, F141 (Fig 4B) is also accessible in the CRD, consistent with the involvement of these two residues in SIRT4 binding as revealed by docking analysis.

At the level of the functional C-RAF-SIRT4 interplay, it is currently unknown whether C-RAF is regulated by an acetylation/deacetylation cycle and whether C-RAF is an enzymatic target of SIRT4. SIRT4 itself exhibits several NAD$^+$-dependent enzymatic activities, including ADP-ribosylation, deacylation, and deacetylation (Betsinger & Cristea, 2019), with recent findings indeed uncovering several new SIRT4 deacetylation targets not only inside, but also outside of the mitochondria (Wang et al, 2022; Zhang et al, 2022). In this context, there is a paradigm for the regulation of B-RAF by SIRT1. Acetylation of B-RAF at lysine 601 by the p300 acetyltransferase promotes B-RAF kinase activity, thereby enhancing the proliferation of melanoma and resistance to B-RAF$^{V600E}$ inhibitors (Dai et al, 2022). On the contrary, SIRT1 deacetylates B-RAF at K601 and therefore inhibits proliferation. Thus, SIRT1 mediates hypoacetylation of B-RAF and therefore (finely) regulates its downstream MAPK signaling activity.

Our results add another layer of complexity to the regulatory network of C-RAF and MAPK signaling by identifying SIRT4 as a C-RAF binder specifically in its inactive state. As summarized in Fig 4E, in mitochondria, SIRT4 inhibits anaplerosis and ultimately ATP generation via inhibition of glutamate dehydrogenase (Haigis et al, 2006). Outside of the mitochondria, SIRT4 interacts, seemingly via its C-terminal R3, with the inactive "closed" form of C-RAF, in which the kinase domain is concealed through 14-3-3 binding to pS259 and pS621. SIRT4 binding to the CRD of C-RAF potentially stabilizes pS259/pS621-C-RAF, thereby preventing membrane recruitment, which is followed by RAS binding and activation of C-RAF.

Consequently, an association of SIRT4 with C-RAF interferes with the activation of downstream MEK/ERK signaling, consistent with findings showing the SIRT4-mediated inhibition of the MAPK pathway (Fu et al, 2017; Chen et al, 2019; Hu et al, 2019; Bai et al, 2020; Tomaselli et al, 2020; Wang et al, 2020).

To date, only the MEK1/2 kinases have been well characterized as substrates of C-RAF. However, there is evidence for kinase-independent functions/activities of C-RAF, including regulation of apoptosis, cell cycle progression, and migration (Nolan et al, 2021). In this context, there is a broad spectrum of C-RAF targets that could interact either directly or indirectly with active (pSer-338) or inactive (pSer-259) forms of C-RAF. This interaction could also be RAS-dependent or RAS-independent. For example, the interaction between MST2 and C-RAF (pSer-259) prevents MST2 dimerization (Romano et al, 2014) and consequently modulates the strength of mitotic and apoptotic signaling. Notably, we also observed an effect of ectopic SIRT4 expression on the Hippo tumor suppressor pathway, which, in addition to the MAPK pathway, also regulates cell proliferation (Ehmer & Sage, 2016; Zinatizadeh et al, 2021). In particular, the increase in pS259-C-RAF levels upon SIRT4 expression (Fig 4C and D) was associated with a decrease in the pYAP/YAP ratio (unpublished results). Taken together, we describe a novel SIRT4-C-RAF axis that negatively affects both MAPK and Hippo-YAP signaling. Another example is ASK1, which normally activates the pro-apoptotic JNK and p38 pathways, and is negatively regulated by C-RAF (Alavi et al, 2007). C-RAF phosphorylated at residue 338 interacts with the N-terminal domain of ASK1 in a kinase-independent and HRAS-dependent manner (Du et al, 2004). The C-RAF-ASK1 complex formed in mitochondria is disrupted by oxidative stress (Matsuzawa et al, 2002). Whether SIRT4 plays a role in this process remains to be investigated. Other C-RAF activities that may be affected by SIRT4 include stimulation of negative regulation of cell migration through direct interaction with ROCKα (Ehrenreiter et al, 2005), promotion of the cell cycle progression through interaction with Polo-like kinase 1 and Aurora kinase A at the mitotic spindle, and the regulation of the DNA damage response through interaction with checkpoint kinase 2 (Mielgo et al, 2011; Zannini et al, 2014; Advani et al, 2015; Joukov & De Nicolo, 2018).

The functional implications of the SIRT4-C-RAF interaction can be extended to apoptosis. Interestingly, C-RAF plays an inhibitory role in mitochondrial apoptosis by promoting BCL-2 and inhibiting BAD (Bajia et al, 2022; Riaud et al, 2024). The latter is characterized by C-RAF–mediated phosphorylation and consequent inactivation of the PKCθ-BAD complex in the control of anti-apoptosis responses (Hindley & Kolch, 2007). In this line, binding of RKIP to C-RAF inhibits its translocation to mitochondria and phosphorylation of BAD, thereby triggering apoptosis as shown in the case of HBx-mediated hepatocarcinogenesis (Kim et al, 2011).

Our study has several limitations. Obtaining structural insights into the effects of the C-RAF$_{CRD}$ mutants in a liquid environment and dynamic system would enhance our understanding of the atomic changes in a more comprehensive manner. However, because of the unavailability of a complete structure of C-RAF (in contrast to B-RAF), we were only able to examine the interactions between SIRT4 and RBD-CRD, and could not address the autoinhibited versus closed conformation of the entire C-RAF protein. Furthermore, targeted inhibition of the SIRT4-C-RAF$_{CRD}$ interaction is required to functionally demonstrate the inhibitory role of SIRT4

overexpression on C-RAF–regulated pathways. This should include both C-RAF kinase–dependent and C-RAF kinase–independent functions, given that C-RAF deficiency causes embryonic lethality in mice (Wojnowski et al, 1998; Huser et al, 2001; Mikula et al, 2001), whereas kinase-deficient C-RAF knock-in mice are viable (Riaud et al, 2024). Therefore, further in-depth characterization of the interaction between SIRT4 and C-RAF$_{CRD}$ at the molecular and cellular/functional levels is required.

# Materials and Methods

## Plasmid constructs

The N-terminal RBD-CRD, RBD, and CRD of RAF kinases were cloned into the pGEX-4T1 vector (BioCat GmbH). Upon transformation into *E. coli*, lysates containing GST-tagged proteins were prepared as previously described (Hemsath & Ahmadian, 2005). The SIRT4 deletion mutants SIRT4(Δ69–98) (Δ1), SIRT4(Δ165–229) (Δ2), and SIRT4(Δ255–314) (Δ3) were generated by PCR-mediated mutagenesis and cloned into pcDNA-3.1 for eukaryotic expression as C-terminal eGFP fusion proteins. The expression construct for N-terminally Flag-tagged C-RAF was kindly provided by Dr. Motta (Genetics and Rare Diseases Research Division, Rome).

## Cell culture and generation of stable cell lines

HEK293 cells were maintained in DMEM serum (Thermo Fisher Scientific) supplemented with 10% FBS (Gibco) and 1% penicillin–streptomycin (Genaxxon). HEK293 cell lines stably expressing eGFP or C-terminally tagged SIRT4-eGFP or SIRT4(ΔN28)-eGFP have been previously described (Lang et al, 2017). In addition, HEK293 cell lines expressing Flag M2 as control or C-terminally Flag M2–tagged SIRT3, SIRT4, or SIRT5 proteins have been described (Bergmann et al, 2020). HEK293 cell lines stably expressing SIRT4(Δ69–98)-eGFP (Δ1), SIRT4(Δ165–229)-eGFP (Δ2), or SIRT4(Δ255–314)-eGFP (Δ3) were generated by transfection using the TurboFect reagent (Thermo Fisher Scientific). Stable HEK293 cell lines were cultured in selection media containing either G418/Geneticin (400 µg/ml; Genaxxon) or puromycin (1.5 µg/ml; Thermo Fisher Scientific). The expression of all SIRT4 constructs was regularly controlled by flow cytometry and/or Western blot analysis.

## Preparing total cell lysates for immunoblot analysis

Cells were lysed on ice for 5 min employing a buffer containing 50 mM Tris–HCl (pH 7.4), 100 mM NaCl, 2 mM MgCl$_2$, 10% glycerol, 20 mM ß-glycerophosphate, 1 mM Na$_3$VO$_4$, 1% IGEPAL (Thermo Fisher Scientific), and 1x protease inhibitor cocktail (Roche). Lysates were cleared by centrifugation (20.000$g$ at 4°C for 5 min). Protein concentrations were determined using the Bradford assay.

## Antibodies for immunoblot analysis

Primary antibodies used in Western blot analysis include anti-GST (own antibody); anti-GFP (1:1,000; #PA1-980; Thermo Fisher

Scientific); anti-Flag (1:1,000; #F742 and #F3165; both from Merck); anti-C-RAF-N-terminal (1:1,000; #ab181115; Abcam); anti-C-RAF-pS259 (1:1,000; #ab173539; Abcam), anti-C-RAF-pY340/341 (1:1,000; #sc-16806; Santa Cruz Biotechnology); anti-vinculin (1:1,000; #V9131; Merck); anti-SIRT4 (1:1,000; #69786; Cell Signaling); anti-ERK(1/2) (1:1,000; #9102; Cell Signaling); anti-p-ERK(1/2) (1:1,000; #4370; Cell Signaling); and anti-KRAS (1:1,000; 11H35L14; Thermo Fisher Scientific). Secondary antibodies employed were from LI-COR (anti-mouse 700 nm: IRDye #926-32213; anti-rabbit 800 nm: IRDye #926-6807).

### Protein purification

The CRD of C-RAF, fused with GST, was cloned individually for each single-point mutation (A142V, L147F, K148T, I154F, Q156R, L160F, N161Q, E174Q, W187Y) and for distinct mutants within Set1 (E174Q, H175R, T178S, K179E, T182L), Set2 (Q156R, F158L, L160F), and Set3 (L147F, K148T, N161Q), using the pGEX-4T1 vector (BioCat GmbH). Fusion proteins were expressed in *E. coli* and subsequently purified using Glutathione High-Capacity Magnetic Agarose Beads (Merck Millipore GmbH) following the manufacturer's guidelines.

### Pull-down assay using GST-fused proteins

Pull-down experiments using GST-fused proteins were performed using glutathione–agarose beads (Macherey-Nagel). The beads were incubated with the GST-fused proteins for 1 h, at 4°C under rotation and centrifuged at 500$g$ followed by three times of washing with ice-cold buffer (30 mM Tris–HCl, 150 mM NaCl, 5 mM MgCl$_2$, and 3 mM DTT). In the next step, samples were incubated with total cell lysates from HEK293 cells stably expressing the indicated Flag-tagged sirtuins or SIRT4-eGFP wild-type and mutants overnight at 4°C followed by three washing steps with ice-cold buffer as indicated above. The protein samples were mixed with Laemmli loading buffer and analyzed by SDS–PAGE and immunoblotting.

### Co-immunoprecipitation analysis

Total cell lysates of HEK293 cells stably expressing C-terminally Flag M2–tagged SIRT4 were incubated overnight at 4°C with anti-Flag M2 agarose beads (Merck). The beads were washed three times with washing buffer (50 mM Tris–HCl, 150 mM NaCl, and 1 mM EDTA). The beads were mixed with Laemmli loading buffer, and co-immunoprecipitation of SIRT4-Flag and endogenous C-RAF proteins was analyzed by SDS–PAGE and immunoblotting. Co-immunoprecipitation of SIRT4-eGFP and endogenous C-RAF using the anti-eGFP nanobody protocol was performed essentially as previously described (Bergmann et al, 2020).

### Densitometric analysis of specific immunoblot protein signals followed by statistical evaluation

Intensities of specific protein bands were determined using Image Studio Lite version 5.2 software. Pull-down data were normalized to the respective total cell lysate signals to ensure an accurate comparison of target protein levels across various samples as previously described (Hsu et al, 2012). Data are presented as the mean ± S.D., and one-way ANOVA statistical evaluation was performed using Origin data analysis software (OriginLab 2021b). Results with at least $P \leq 0.05$ were considered significant (* $P \leq 0.05$; ** $P \leq 0.01$; and *** $P \leq 0.001$).

### Structural analysis

We created a structural homology model of human SIRT4 based on the X-ray diffraction structure of SIRT4 from *X. tropicalis* (PDB: 5OJ7) and compared it with human SIRT5 (PDB: 4G1C) for mutational analysis using PyMOL (version 4.6.0). Moreover, because of the absence of a complete structure of inactive C-RAF, we employed a comparative approach by superimposing the structures of inactive B-RAF (full-length; PDB: 6NYB) to gain insights into the potential structure of inactive C-RAF (Park et al, 2019) in complex with 14-3-3. The three-dimensional structure of the resulting inactive state of C-RAF was analyzed and visualized using PyMOL (version 4.6.0).

### Molecular docking simulations

The crystal structures of the C-RAF$_{CRD}$ (PDB: 1FAQ) and KRAS-C-RAF$_{RBD-CRD}$ complex (PDB: 6XHB) were obtained from the Protein Data Bank (PDB), and the human full-length SIRT4(AF-Q9Y6E7) structure was obtained from the AlphaFold database (https://alphafold.ebi.ac.uk/). Molecular docking simulations were performed using default mode settings available in the molecular docking ClusPro 2.0 server (Kozakov et al, 2017). From the refined selection of proposed structures, a configuration exhibiting optimal binding energies was chosen, aligning it with experimental data. After the docking simulations, the resulting structures were meticulously examined to identify significant molecular interactions using BIOVIA software.

# Supplementary Information

# Acknowledgements

We are grateful to our colleagues from the Institute of Biochemistry and Molecular Biology II for their support, helpful advice, and fruitful discussions. We thank Dr. Motta (Genetics and Rare Diseases Research Division, Rome) for providing the Flag-C-RAF expression vector. This study was supported by the German Research Foundation (DFG; grant AH 92/8-3 to MR Ahmadian), by the European Network on Noonan Syndrome and Related Disorders (NSEuroNet; grant 01GM1602B to MR Ahmadian), and in part by the Foundation for Ageing Research of the Heinrich Heine University (grants 701.810.783 to RP Piekorz, and 701.810.845 to MR Ahmadian).

### Author Contributions

M Mehrabipour: conceptualization, data curation, software, formal analysis, validation, investigation, visualization, methodology, and writing—original draft, review, and editing.

S Nakhaei-Rad: software, formal analysis, investigation, methodology, and writing—review and editing.

R Dvorsky: software, formal analysis, investigation, methodology, and writing—review and editing.

A Lang: resources, investigation, methodology, and writing—review and editing.

P Verhülsdonk: formal analysis, investigation, and methodology.

MR Ahmadian: conceptualization, resources, data curation, formal analysis, supervision, funding acquisition, validation, investigation, visualization, methodology, project administration, and writing—original draft, review, and editing.

RP Piekorz: conceptualization, resources, data curation, formal analysis, supervision, funding acquisition, validation, investigation, visualization, methodology, project administration, and writing—original draft, review, and editing.

## Conflict of Interest Statement

The authors declare that they have no conflict of interest.

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
