## [Reviewer comments · Life Science Alliance]

Life Science Alliance

SIRT4 as a novel interactor and candidate suppressor of C-RAF kinase in MAPK signaling

Mehrnaz Mehrabipour, Saeideh Nakhaei-Rad, Radovan Dvorsky, Alexander Lang, Patrick Verhuelsdonk, Mohammad Ahmadian, and Roland Piekorz

DOI: <https://doi.org/10.26508/lsa.202302507>

Corresponding author(s): Roland Piekorz, Institute of Biochemistry and Molecular Biology II, Medical Faculty and University Hospital Duesseldorf, Heinrich Heine University, 40225 Duesseldorf, Germany and Mohammad Ahmadian, Heinrich-Heine University

Review Timeline:

Submission Date:	2023-12-04
Editorial Decision:	2024-01-26
Revision Received:	2024-02-29
Editorial Decision:	2024-03-05
Revision Received:	2024-03-06
Accepted:	2024-03-07

Transaction Report:

January 26, 2024

Re: Life Science Alliance manuscript #LSA-2023-02507-T

Dr. Roland P Piekorz

Institute of Biochemistry and Molecular Biology II, Medical Faculty and University Hospital Duesseldorf, Heinrich Heine University, 40225 Duesseldorf, Germany

Dear Dr. Piekorz,

Thank you for submitting your manuscript entitled "SIRT4 as a novel interactor and candidate suppressor of C-RAF kinase in MAPK signaling" to Life Science Alliance. The manuscript was assessed by expert reviewers, whose comments are appended to this letter. We invite you to submit a revised manuscript addressing the Reviewer comments.

Thank you for this interesting contribution to Life Science Alliance. We are looking forward to receiving your revised manuscript.

Sincerely,

B. MANUSCRIPT ORGANIZATION AND FORMATTING:

Reviewer #1 (Comments to the Authors (Required)):

This is an excellent work that I think the readership will find of interest. SIRT4 is a mitochondrial tumor suppressor protein. If I understand correctly, what the authors are suggesting is that SIRT4 hijacks the CRD of Raf-1 (C-RAF), preventing Raf-1 from localizing at the membrane. This is a significant mechanistic observation since it can explain how SIRT4 suppresses MAPK signaling.

Although not in the mitochondria, and differing functionally too, mechanistically it resembles RAP1 mode of action (PMID: 32249186), where the authors proposed that Rap1 suppresses MAPK signaling via Raf-1 (C-RAF) but not via BRAF with the suggested mechanism being that Rap1 high-affinity binding to Raf-1's cysteine-rich domain (CRD) disfavors Rap1 interaction with Raf-1's Ras-binding domain (RBD). So here CRD is hijacked by RAP1.

Overall, a coherent mechanistic story involving an important system. The work is comprehensive, and the writing and figures are clear.

Reviewer #2 (Comments to the Authors (Required)):

The study investigates the molecular and functional interaction between the proto-oncogene C-RAF and the tumor suppressor SIRT4 in the context of MAPK signaling inhibition. C-RAF is a key kinase in the MAPK pathway, and SIRT4 is a sirtuin involved in metabolic regulation. The authors identify a novel interaction between C-RAF and SIRT4, showing that SIRT4 selectively binds to the N-terminal CRD domain of C-RAF. Mutational analysis reveals specific residues crucial for this interaction. Importantly, SIRT4 interacts with the inactive form of C-RAF, inhibiting MAPK signaling and cell proliferation. The study suggests a novel role for SIRT4 in regulating the MAPK pathway through its interaction with C-RAF.

1. The molecular docking simulations and mutational analysis provide valuable insights into the potential binding interface between C-RAF_{CRD} and SIRT4. However, the significance of gain-of-function mutations in C-RAF_{CRD} could be discussed in more detail.
2. The discussion highlights the potential implications of the SIRT4-C-RAF interaction in inhibiting the MAPK signaling pathway. However, the text could delve deeper into the functional consequences and broader implications of this interaction.
3. The limitations of the study and potential areas for future research could be discussed to provide a more balanced view of the findings.

Point-by-point reply

Comment to the editor and reviewers:

Unfortunately, in Fig. 1F (Repeat 1; Vinculin immunoblot), Fig. S5C (Repeat 3; middle panel, GST-C-RAF_{Nterm} immunoblot), and Fig. S8A (Repeat 2, lower panel, Vinculin immunoblot), errors (wrong blots) were included in the preparation of the figure panels. Please accept our sincere apologies for these errors, which are now corrected in the revised version with the correct immunoblot figures. Additionally, we have followed LSA's suggestion to provide all immunoblot data from the main and supplementary figures as original uncropped images (SourceDataForFigures, a total of 11 files provided).

Reviewer #1:

This is an excellent work that I think the readership will find of interest. SIRT4 is a mitochondrial tumor suppressor protein. If I understand correctly, what the authors are suggesting is that SIRT4 hijacks the CRD of Raf-1 (C-RAF), preventing Raf-1 from localizing at the membrane. This is a significant mechanistic observation since it can explain how SIRT4 suppresses MAPK signaling.

Although not in the mitochondria, and differing functionally too, mechanistically it resembles RAP1 mode of action (PMID: 32249186), where the authors proposed that Rap1 suppresses MAPK signaling via Raf-1 (C-RAF) but not via BRAF with the suggested mechanism being that Rap1 high-affinity binding to Raf-1's cysteine-rich domain (CRD) disfavors Rap1 interaction with Raf-1's Ras-binding domain (RBD). So here CRD is hijacked by RAP1.

Overall, a coherent mechanistic story involving an important system. The work is comprehensive, and the writing and figures are clear.

Response to the reviewer:

We thank the reviewer for the positive and encouraging feedback as well as for bringing in the very interesting link between CRAF-CRD and RAP1. This functional interaction was incorporated as follows in the discussion part of the paper (p. 6/7):

Interestingly, analogous to our finding, the role of C-RAF_{CRD} interaction in an isoform-specific manner with another C-RAF regulator to inhibit the MAPK pathway has been demonstrated for RAP1 (Nussinov et al, 2020). Here, RAP1 inhibits MAPK signaling via interaction with C-RAF_{CRD} by reducing the number of clustered oncogenic Ras molecules, thereby suppressing C-RAF (but not B-RAF) activation and MAPK signaling. The presence of RAP1 within the nanoclusters competes with RAS for C-RAF as a common target, resulting in the suppression of C-RAF activation. However, whereas RAP1 interacts with the open form of C-RAF at the cell membrane, our data suggest that SIRT4 binds to the autoinhibited (closed) form of C-RAF. Regardless, similar to RAP1, SIRT4 may functionally hijack and inhibit C-RAF via its CRD domain.

Reviewer #2:

The study investigates the molecular and functional interaction between the proto-oncogene C-RAF and the tumor suppressor SIRT4 in the context of MAPK signaling inhibition. C-RAF is a key kinase in the MAPK pathway, and SIRT4 is a sirtuin involved in metabolic regulation. The authors identify a novel interaction between C-RAF and SIRT4, showing that SIRT4 selectively binds to the N-terminal CRD domain of C-RAF. Mutational analysis reveals specific residues crucial for this interaction. Importantly, SIRT4 interacts with the inactive form of C-RAF, inhibiting MAPK signaling and cell proliferation. The study suggests a novel role for SIRT4 in regulating the MAPK pathway through its interaction with C-RAF.

1. The molecular docking simulations and mutational analysis provide valuable insights into the potential binding interface between C-RAF_{CRD} and SIRT4. However, the significance of gain-of-function mutations in C-RAF_{CRD} could be discussed in more detail.

Response to the reviewer:

To obtain more detailed information about the gain-of-function (GOF) mutations in C-RAF_{CRD}, we generated mutants *in silico* and molecular docking analysis was conducted. Comparing the wild-type C-RAF_{CRD} interaction with SIRT4 with the mutated Set1 and Set2 variants of C-RAF_{CRD}, we observed that Set 1 and Set 2 have lower, more stable cluster scores. We have included a supplementary Excel file as **Table S2** and **Fig. S7** for your reference, which will assist in elucidating and comparing the binding sites and in understanding the potential restructuring of the binding interface and increased stability of the C-RAF_{CRD} -SIRT4 interaction in the setting of the Set1 and Set2 mutations. This information has been included in the "Mutational Analysis of the C-RAF_{CRD} and SIRT4 Interaction" results section of the manuscript (p. 4/5):

To identify residues of the C-RAF_{CRD} - SIRT4 binding interface and obtain a more detailed insight into their intermolecular interplay, we performed molecular docking analysis between C-RAF_{CRD} (PDB: 1FAQ) and full-length SIRT4 (Q9Y6E7) using the ClusPro 2.0 server. The 3D surface structure (**Fig. 3G**) highlights the binding between C-RAF_{CRD} and R3 of SIRT4, along with certain parts of R1. For a more detailed understanding of this intermolecular binding, analysis of the binding surface utilizing the BIOVIA software revealed an interacting network (**Fig. 3H**), in which the stability of the C-RAF_{CRD} - SIRT4 complex is the result of a combination of various interaction types, i.e., hydrogen bonds, electrostatic interactions, and hydrophobic contacts (**Table S1**). For example, the C-RAF_{CRD} residue K157 and the SIRT4 residue D236 form a hydrogen/electrostatic bond with a distance of 1.8 Å, indicative of a strong interaction. C-RAF_{CRD} residues R143, K157, H175, T178, K179, Q156, E174, S177, N161, I154, and SIRT4 residues R75, R97, T274, H92, T237, D236, Q264, Q91, R270, R291, G93, G235, and Y266 further contribute to the binding stability via hydrogen bonds. Notably, electrostatic interactions were observed between C-RAF_{CRD} residues R143, E174, and F141 and SIRT4 residues E277, R270, and R291, respectively (**Fig. 3H; Table S1**). Moreover, hydrophobic interactions were identified involving residues of C-RAF_{CRD} (H175, L160, F163, R143) and SIRT4 (V232, F234, P240, Y266, R270).

Because the C-RAF_{CRD} Set1 and Set2 mutations resulted in stronger binding to SIRT4-Flag (**Fig. 3C-F**), further molecular docking analysis was performed for these C-RAF_{CRD} gain-of-function (GOF) mutations. Comparing the cluster scores of wild-type C-RAF_{CRD} interacting with SIRT4 shows a weighted score of -716 for both the middle and lowest energy. In contrast, Set1 and Set2 have lower, more stable cluster scores: -738.7 and -795 for the center and the lowest energy in the case of Set1, and -744 for both the center and the lowest energy in the case of Set2. The combined mutations in Set1, particularly the E174Q, H175R, T175S, K179E, and T182L mutations, alter the interaction profile of C-RAF_{CRD} with SIRT4, thereby forming new hydrogen bonds as well as electrostatic and hydrophobic contacts, which potentially enhance complex stability (**Fig. S7D** and **Table S2**). Although some interactions are lost in Set1 compared to wild-type C-RAF_{CRD} (**Table S2**), considering the cluster score and the mode of binding, we propose also new platforms of interactions. These involve a new set of C-RAF_{CRD} residues, i.e., D153, Y170, P181, L182, M183, and V185, that might collectively increase the binding affinity of Set1 with SIRT4 (**Fig. S7D** and **Table S2**). Moreover, compared to wild-type C-RAF_{CRD}, the mutations within Set1 induce a modified interaction pattern with SIRT4, characterized by increased interaction of C-RAF_{CRD} residues with R1 of SIRT4 while exhibiting reduced interaction with R3 and the SIRT4 gray area (which lacks R1, R2, and R3) (**Fig. S7A** and **S7C**).

Similarly to C-RAF_{CRDSet1}, Set2 mutations in the C-RAF_{CRD} region also introduce new interactions, as well as changes in the type and distance of existing interactions with the respective SIRT4 regions (**Fig. S7E and F**). For instance, the F158L mutation leads to the formation of a new hydrogen bond with T237 of SIRT4, and the L160F mutation results in the interaction with both P240 and V243 of SIRT4, leading to a higher involvement of CRD-Set2 residues (**Fig. S7F and Table S2**). Notably, in the case of C-RAF_{CRDSet2}, the C-RAF_{CRD} residues C155, L158, F172, and H173, undergo novel hydrogen bonds with SIRT4 residues, suggesting a restructuring of the binding interface and thereby increasing the stability of the C-RAF_{CRD}-SIRT4 interaction in the case of C-RAF_{CRDSet2} (**Fig. S7F and Table S2**).

2. The discussion highlights the potential implications of the SIRT4-C-RAF interaction in inhibiting the MAPK signaling pathway. However, the text could delve deeper into the functional consequences and broader implications of this interaction.

Response to the reviewer:

Thanks for this advice. We did expand the discussion towards more functional implications of the SIRT4-C-RAF interaction in the discussion (p. 8):

The functional implications of the SIRT4 -C-RAF interaction can be extended to apoptosis. Interestingly, C-RAF plays an inhibitory role in mitochondrial apoptosis by promoting BCL-2 and inhibiting BAD (Bajja et al, 2022; Riaud et al, 2024). The latter is characterized by C-RAF-mediated phosphorylation and consequent inactivation of the PKC θ -BAD complex in the control of anti-apoptosis responses (Hindley & Kolch, 2007). In this line, binding of RKIP to C-RAF inhibits its translocation to mitochondria and phosphorylation of BAD, thereby triggering apoptosis as shown in the case of HBx-mediated hepato-carcinogenesis (Kim et al, 2011).

3. The limitations of the study and potential areas for future research could be discussed to provide a more balanced view of the findings.

Response to the reviewer:

In fact, we agree with the reviewer that the manuscript needs to be balanced in a way that both highlights the limitations of this study and provides further steps. This has been incorporated as follows in the discussion part of the paper (p. 8):

Our study has several limitations. Obtaining structural insights into the effects of the C-RAF_{CRD} mutants in a liquid environment and dynamic system would enhance our understanding of the atomic changes in a more comprehensive manner. However, due to the unavailability of a complete structure of C-RAF (in contrast to B-RAF), we were only able to examine the interactions between SIRT4 and RBD-CRD, and could not address the auto-inhibited vs. closed conformation of the entire C-RAF protein. Furthermore, targeted inhibition of the SIRT4-C-RAF_{CRD} interaction is required to functionally demonstrate the inhibitory role of SIRT4 overexpression on C-RAF regulated pathways. This should include both C-RAF kinase-dependent and -independent functions, given that C-RAF deficiency causes embryonic lethality in mice (Wojnowski et al, 1998; Huser et al, 2001; Mikula et al, 2001), whereas kinase-deficient C-RAF knock-in mice are viable (Riaud et al, 2024). Therefore, further in-depth characterization of the interaction between SIRT4 and C-RAF_{CRD} at the molecular and cellular/functional levels is required.

Literature:

- Bajja D, Bottani E, Derwich K (2022) Effects of noonan syndrome-germline mutations on mitochondria and energy metabolism. *Cells* 11: doi:10.3390/cells11193099
- Hindley A, Kolch W (2007) Raf-1 and b-raf promote protein kinase c theta interaction with bad. *Cell Signal* 19: 547-555. doi:10.1016/j.cellsig.2006.08.004
- Huser M, Luckett J, Chiloeches A, Mercer K, Iwobi M, Giblett S, Sun XM, Brown J, Marais R, Pritchard C (2001) Mek kinase activity is not necessary for raf-1 function. *EMBO J* 20: 1940-1951. doi:10.1093/emboj/20.8.1940
- Kim SY, Park SG, Jung H, Chi SW, Yu DY, Lee SC, Bae KH (2011) Rkip downregulation induces the hbx-mediated raf-1 mitochondrial translocation. *J Microbiol Biotechnol* 21: 525-528. doi:10.4014/jmb.1012.12023
- Mikula M, Schreiber M, Husak Z, Kucerova L, Ruth J, Wieser R, Zatloukal K, Beug H, Wagner EF, Baccarini M (2001) Embryonic lethality and fetal liver apoptosis in mice lacking the c-raf-1 gene. *EMBO J* 20: 1952-1962. doi:10.1093/emboj/20.8.1952
- Nussinov R, Jang H, Zhang M, Tsai CJ, Sablina AA (2020) The mystery of rap1 suppression of oncogenic ras. *Trends Cancer* 6: 369-379. doi:10.1016/j.trecan.2020.02.002
- Riaud M, Maxwell J, Soria-Bretones I, Dankner M, Li M, Rose AAN (2024) The role of craf in cancer progression: From molecular mechanisms to precision therapies. *Nat Rev Cancer* 24: 105-122. doi:10.1038/s41568-023-00650-x
- Wojnowski L, Stancato LF, Zimmer AM, Hahn H, Beck TW, Larner AC, Rapp UR, Zimmer A (1998) Craf-1 protein kinase is essential for mouse development. *Mech Dev* 76: 141-149. doi:10.1016/s0925-4773(98)00111-7

March 5, 2024

RE: Life Science Alliance Manuscript #LSA-2023-02507-TR

Dr. Roland P Piekorz

Institute of Biochemistry and Molecular Biology II, Medical Faculty and University Hospital Duesseldorf, Heinrich Heine University, 40225 Duesseldorf, Germany

Dear Dr. Piekorz,

Thank you for submitting your revised manuscript entitled "SIRT4 as a novel interactor and candidate suppressor of C-RAF kinase in MAPK signaling". We would be happy to publish your paper in Life Science Alliance pending final revisions necessary to meet our formatting guidelines.

- please be sure that the authorship listing and order is correct
- please add ORCID ID for the secondary corresponding author -- they should have received instructions on how to do so
- please add the Twitter handle of your host institute/organization as well as your own or/and one of the authors in our system
- please upload your Tables in editable .doc or Excel format
- please add callouts for Figures S4A, B; S5A-C; S6A,B; S7B to your main manuscript text

A. FINAL FILES:

B. MANUSCRIPT ORGANIZATION AND FORMATTING:

****It is Life Science Alliance policy that if requested, original data images must be made available to the editors. Failure to provide**

original images upon request will result in unavoidable delays in publication. Please ensure that you have access to all original data images prior to final submission.**

The license to publish form must be signed before your manuscript can be sent to production. A link to the electronic license to publish form will be available to the corresponding author only. Please take a moment to check your funder requirements.

Sincerely,

March 7, 2024

RE: Life Science Alliance Manuscript #LSA-2023-02507-TRR

Dr. Roland P Piekorz
Institute of Biochemistry and Molecular Biology II, Medical Faculty and University Hospital Duesseldorf, Heinrich Heine
University, 40225 Duesseldorf, Germany

Dear Dr. Piekorz,

Thank you for submitting your Research Article entitled "SIRT4 as a novel interactor and candidate suppressor of C-RAF kinase in MAPK signaling". It is a pleasure to let you know that your manuscript is now accepted for publication in Life Science Alliance. Congratulations on this interesting work.

DISTRIBUTION OF MATERIALS:

Again, congratulations on a very nice paper. I hope you found the review process to be constructive and are pleased with how the manuscript was handled editorially. We look forward to future exciting submissions from your lab.

Sincerely,
